# Oscillation of Sympathetic Activity in Patients with Obstructive Sleep Apnea during the First Hour of Sleep

**DOI:** 10.3390/healthcare11192701

**Published:** 2023-10-09

**Authors:** Jui-Kun Chiang, Yen-Chang Lin, Yee-Hsin Kao

**Affiliations:** 1Department of Family Medicine, Dalin Tzu Chi Hospital, Buddhist Tzu Chi Medical Foundation, No. 2, Minsheng Road, Dalin, Chiayi 622, Taiwan; roma@tzuchi.com.tw; 2Nature Dental Clinic, Puli Township, Nantou 404, Taiwan; drlin@alliswell.tw; 3Department of Family Medicine, Tainan Municipal Hospital (Managed by Show Chwan Medical Care Corporation), 670 Chung-Te Road, Tainan 701, Taiwan

**Keywords:** heart rate variability (HRV), apnea/hypopnea index (AHI), obstructive sleep apnea (OSA), low frequency/high frequency (LF/HF) ratio, root mean square of successive differences (RMSSD)

## Abstract

(1) Background: Snoring is a cardinal symptom of obstructive sleep apnea (OSA) and has been suggested to potentially increase sympathetic activity. On the other hand, sleep itself usually leads to a decrease in sympathetic activity. Heart rate variability (HRV) analysis is a non-invasive technique used to assess autonomic nervous system function. However, there is limited research on the combined impact of sleep and snoring on sympathetic activity in individuals with OSA, particularly during the first hour of sleep (non-rapid eye movement sleep). The current study aims to investigate the net effect of sleep and snoring on sympathetic activity and explore factors that might contribute to increased sympathetic activity in individuals with OSA during the first hour of sleep. (2) Methods: The participants were referred from the outpatient department for OSA diagnosis and underwent whole-night polysomnography (PSG). Electrocardiogram (EKG) data from the PSG were downloaded for HRV analysis. HRV measurements were conducted in both the time and frequency domain, including the root mean square of successive differences between normal heartbeats (RMSSD) and the ratio of the absolute power of the low-frequency (LF) band (0.04–0.15 Hz) to the absolute power of the high-frequency (HF) band (0.15–0.4 Hz) (LF/HF ratio), respectively. (3) Results: A total of 45 participants (38 men and 7 women) were included in the analysis. The RMSSD gradually increased from 0–5 min to 50–60 min (*p* = 0.024), while the LF/HF ratio decreased (*p* < 0.001) during the first hour of sleep (non-rapid eye movement sleep). The LF/HF ratios of the “S” (snoring) episodes were compared with those of the pre-S episodes. An elevated LF/HF ratio during the S episode was associated with the first snoring episode occurring more than 20 min after lying down to sleep (Odds ratio, OR = 10.9, *p* = 0.004) and with patients diagnosed with severe OSA (OR = 5.01, *p* = 0.045), as determined by logistic regression. (4) Conclusions: The study observed an increase in the value of RMSSD and a decrease in the value of the LF/HF ratio during the first hour of sleep for patients with OSA. Higher LF/HF ratios were associated with the first occurrence of snoring while lying down for more than 20 min and with patients with severe OSA.

## 1. Introduction

Sleep is important to a person’s overall health, including physical, mental, and social well-being. In normal individuals, the transition from wakefulness to non-rapid eye movement (non-REM) sleep is accompanied by an increase in parasympathetic drive and a decrease in sympathetic activity, promoting a more relaxed and restorative state during sleep [1]. However, during REM sleep, there is an increase in sympathetic activity in normal subjects, which is possibly linked to changes in muscle tone [2]. Obstructive sleep apnea (OSA) is the most common sleep breathing disorder, with a prevalence ranging from 9% to 38% in the general population [3]. The incidence of OSA has increased over the past two decades [4]. Obstructive sleep apnea has emerged as a prominent, yet largely treatable, contributor to various cardiovascular diseases [5]. OSA is characterized by recurrent episodes of partial or complete upper airway collapse during sleep, resulting in hypopnea or apnea lasting for more than 10 s. Furthermore, it is associated with either cortical arousal or a fall in blood oxygen saturation [6]. Previous research has shown that excessive sleepiness, fatigue, or unrefreshing sleep (73–90%) is the most common presenting symptom of OSA, followed by snoring (50–60%) [6]. On the other hand, in patients with OSA, the pattern of sympathetic activity during sleep can be different.

Snoring is a common sleep-related condition characterized by the rough, hoarse, or harsh noise produced during sleep due to the vibration of soft tissues in the upper airway. It affects a larger proportion of men (40%) compared to women (20%) in the general population [7,8,9]. Epidemiological studies have indicated that snoring is a significant risk factor for hypertension and cardiovascular disease [10,11,12], and the likelihood of developing hypertension increases with the loudness of snoring [13]. While the underlying pathophysiology is complex and multifactorial, sympathetic nervous system activation is believed to be involved in the increased cardiovascular risk seen in OSA [14]. Previous studies have reported that snoring could increase sympathetic activity. These findings were based on alterations in breathing frequency, inspiratory and expiratory time, and tidal volume observed in response to snoring or simulated snoring using high-frequency oscillations [15,16,17]. These changes suggest an impact on the autonomic nervous system, with an increase in sympathetic nervous system activity during snoring episodes. 

Polysomnography (PSG) is considered the gold standard method for diagnosing OSA and monitoring snoring [18]. By utilizing a reliable algorithm based on the typical features of snoring frequency and amplitude, the exact snoring segment can be calculated, allowing for visualization of the snoring segment on a spectrogram [19]. On the other hand, heart rate variability (HRV) serves as a non-invasive method to assess the regulation of the autonomic nervous system. A systematic review revealed that adult patients with OSA exhibited heightened sympathetic responsiveness [20]. Among the frequency domain variables, the low frequency/high frequency (LF/HF) ratio is regarded as an index of sympathovagal balance [21]. A previous study showed that longer recording epochs better represent processes with slower fluctuations and the cardiovascular system’s response to a wider range of environmental stimuli and workloads [22]. It is difficult to use the average value of HRV in the whole night to distinguish and compare the HRV of epochs before and during snoring events. The reasons for choosing 5 or 10 min epochs for recording might include considerations regarding the timing of first snoring and duration of snoring in patients with OSA. Therefore, this study aims to search for the first snoring event and compare the HRV of the epoch before snoring and the first snoring during the first hour of sleep. The current study aims to investigate the net effect of sleep and snoring on sympathetic activity and explore factors that might contribute to increased sympathetic activity in individuals with OSA during the first hour of sleep (non-REM sleep).

## 2. Materials and Methods

### 2.1. Ethics

Informed consent was obtained from all patients prior to their enrollment in the study. The study protocol was reviewed and approved by the institutional review board of the Tainan Municipal Hospital (managed by Show Chwan Medical Care Corporation) (SCMH_IRB No: 1090508).

### 2.2. Study Participants

A total of 57 consecutive participants who underwent PSG for clinically suspected OSA and were referred to the sleep unit of a teaching hospital in southern Taiwan were recruited. Most patients were referred from otorhinolaryngology or internal medical department in the same hospital during July 2020 to June 2021. The inclusion criteria were patients receiving nocturnal PSG with apnea/hypopnea index (AHI) ≥ 5 and snoring in the first hour of sleep. Participants with no snoring all night (n = 3), snoring 1 h after sleeping (n = 4), arrhythmia (n = 1), snoring within 3 min after lying down (n = 1), and AHI < 5 (n = 3) were excluded. All subjects did not take medications that affect the sympathetic nervous system (e.g., beta-blockers, alpha-blockers, and centrally acting drugs).

### 2.3. PSG and HRV Measurement

Participants underwent overnight PSG (EMBLA N7000 system, Embla Inc., Broomfield, CO, USA). The PSG collects electrophysiological signals for heart activity analysis, pulse oximetry readings, airflow by using nasal pressure and oronasal thermal sensors, body position, actigraphy data, and thoracic and abdominal movements [23]. The electrocardiogram (EKG) signals from the PSG were downloaded for HRV analysis. Time- and frequency-domain analyses were performed. For time domain analysis, the time period for analysis was chosen. The root mean square of successive differences (RMSSD), a time-domain measurement index, was recognized as a marker of parasympathetic activity [24]. The frequency-domain HRV analysis was performed by applying the Fast Fourier Transform. The frequency analysis window was 300 s and shifted by 10 s. The values are ultra-low-frequency component (ULF) = [0, 0.03] Hertz (Hz), very low-frequency component (VLF) = [0.03, 0.05] Hz, low-frequency component (LF) = [0.05, 0.15] Hz, and high-frequency component (HF) = [0.15, 0.4] Hz. The mean values for the above components were calculated from the selected durations [either 5 min or 10 min] [25]. One of the frequency-domain measurement indices used was the ratio between the absolute power of the HF band and the absolute power of the LF band, referred to as the LF/HF ratio. The LF/HF ratio is considered a measure of the sympathovagal balance index [21]. In the current study, the ratio of low-frequency power to high-frequency power (LF/HF ratio) was utilized to measure sympathetic activity, as referenced in the study by AlQatari et al. [26]. 

The matrix correlation for the parameters of the frequency domain was performed. The times of PSG and snoring sounds were matched. The first 60 min of sleep were selected for the analysis to lie in the non-REM stage and to avoid possible motion interferences during the whole-night sleep period. The mean HRV was evaluated every 10 min. The centers of the episodes were the 5th, 15th, 25th … to 55th minutes. In this study, the episode in which the initial snoring was observed was defined as the “S episode”, and the episode preceding the S episode was defined as the “pre-S episode”, as referenced in a previous study [19]. Once the S episode and pre-S episode were chosen, the corresponding mean HRV function could be defined. The mean HRV indexes were calculated for every episode. Data files were visually inspected for artifacts [23].

In the current study, the change of LF/HF ratio from pre-S episodes to S episodes for the first snoring in patients with OSA was analyzed, and then stratified into two groups based on the differences between pre-S and S episodes: the increased group and the decreased group. HRV measurements during the pre-snoring (non-snoring) and snoring periods were accurately obtained using a reliable algorithm that can detect the exact onset of snoring. This visualizing algorithm of snoring revealed that the typical features of the clustered snores in the amplitude domains were near-isometric spikes (most had an ascending–descending trend) and had one or more visibly fixed frequency bands [19]. 

### 2.4. Statistical Analysis

Continuous variables were presented as mean ± standard deviation where t-test, paired t-test, or Wilcoxon rank sum were indicated by cases. The continuous data were presented as mean ± S.D. or median (interquartile range) as indicated. Categorical variables were presented as n (%) where the chi-square test was applied. Hilbert transformation and analysis using R with the ebm, seewave, pracma, and RHRV packages were for the HRV. One-way repeated measures analysis of variance (ANOVA) was performed to evaluate the differences between groups. Time series plots were generated. All factors listed in Table 1 and Table 2 were included during the logistic regression analysis. Stepwise factor selection (both forward and backward) was performed. Model diagnostics were also performed for the final model. Data entry and analysis were performed using the free R software, version 4.0.3 (R Foundation for Statistical Computing, Vienna, Austria). All statistical assessments were two-sided, and statistical significance was set at the 0.05 level.

## 3. Results

A total of 57 participants were collected during the study period. A total of 45 participants (38 men and 7 women) were enrolled for analysis in the current study. The demographic data for participants are presented in Table 1. 

This current study found that the median (interquartile range) AHI (apnea/hypopnea index) was 20.6 (9.1, 56.6), and was significantly higher in the severe OSA group (*p* < 0.001). The centers of the pre-S episodes and S episodes were both significantly higher in the mild–moderate OSA group than in the severe OSA group (*p* = 0.019; *p* = 0.016). HRV was analyzed at different time intervals after lying down (0–5 min, 5–10 min, 10–20 min, 20–30 min, 30–40 min, 40–50 min, and 50–60 min), and the timing of initial snoring was confirmed through the method described in a previous study, which involved a visualizing snoring method [18]. In the current study, the first occurrence of snoring in all participants was detected during the non-rapid eye movement (non-REM) sleep stage. 

Ten participants were randomly selected to display the individual values of LF/HF ratio in Figure 1 and the individual values of RMSSD in Figure 2.

The LF/HF (low frequency/high frequency) ratio and RMSSD (root mean square of successive differences) of the S episodes and pre-S episodes were not significantly different between mild–moderate OSA and the severe OSA group, as shown in Table 1. The RMSSD gradually increased from 0–10 min to 50–60 min (*p* = 0.059), while the LF/HF ratios had a trend of decrease from 0–10 min to 50–60 min (*p* = 0.100) (Table 2). The LF/HF ratio had a significantly positive correlation with LF and a negative correlation with HF. This LF/HF ratio was used as the indicator of sympathetic activity [27]. Given the higher variability of the LF/HF ratio within the first 10 min of sleep in the current study (Table 2), we divided it into two segments: 0–5 min and 5–10 min for further analysis. We found that the LF/HF ratio significantly decreased during the first 20 min after sleep (*p* < 0.001).

For better illustration, the ratios of LF/HF were plotted for each case during the first hour of sleep. The LF/HF ratios of the pre-S episodes with those of the S episodes comprising the first snoring were compared and then stratified into two groups based on the differences between the pre-S and S episodes: the increased group and the decreased group (Figure 3). The star points in Figure 3 represented centers of pre-S episodes, while the black round points represented centers of S episodes. The analysis showed that the LF/HF ratios increased in 25 (55.6%) patients and decreased in 20 (44.4%) patients (Table 3). The initial LF/HF ratio within 5 min after lying down in bed was not significantly different between these two groups. The median center of the S episode for the increased group was larger than that of the decreased group, although it did not reach statistical significance: 25.0 (7.5, 35.0) vs. 15.0 (7.5, 15.0), respectively (*p* = 0.216). The median center of the pre-S episodes was also higher in the increased group than in the decreased group, although it did not reach statistical significance (15 (2.5, 25) vs. 7.5 (2.5, 7.5), *p* = 0.134) (Table 3). Among the decreased group, the LF/HF ratio decreased from 2.4 ± 1.0 (pre-S episode) to 1.5 ± 0.9 (S episode) (*p* < 0.001). Conversely, the LF/HF ratio significantly increased from 1.4 ± 0.8 (pre-S episode) to 2.0 ± 1.0 (S episode) in the increased group (*p* < 0.001). Logistic regression analysis revealed a higher probability of an increased LF/HF ratio that was significantly associated with initial snoring of lying down for more than 20 min in patients with OSA (OR = 10.9, *p* = 0.004), and patients with severe OSA (OR = 5.01, *p* = 0.045) (Table 4). The Hosmer–Lemeshow test was passed (*p* = 0.83). The discrimination power for this fitted logistic regression model was 0.779 with 95% CI: 0.642–0.916.

## 4. Discussion

In the current study, light was shed on the relationship between the LF/HF ratios during the first snoring episode and specific sleep-related parameters, emphasizing the potential clinical significance of autonomic nervous system (ANS) activity in the context of snoring and sleep disorders. The observation revealed that the value of RMSSD increased while the value of the LF/HF ratio decreased during the first hour of lying down to sleep for patients with OSA. Further observations indicated that a higher probability of an increased LF/HF ratio was associated with the first occurrence of snoring while lying down for more than 20 min and with patients with severe OSA.

A previous study reviewed HRV in adult OSA patients and reported that people with OSA exhibit reduced vagal tone and higher sympathetic sensitivity [20]. Another study reported that non-REM sleep (sleep onset) was associated with increased parasympathetic activity, whereas REM sleep was associated with a shift toward greater sympathetic modulation for healthy young adults [25,26]. In the current study, an increase in parasympathetic activity but a decrease in sympathetic activity during the first hour of sleep was found for patients with OSA. Although the timing and duration of REM sleep can vary depending on a person’s individual sleep patterns and other factors, rapid eye movement sleep usually occurs about 90 min after the onset of sleep [2]. 

Furthermore, the change of the LF/HF ratio from pre-S episodes to S episodes was analyzed for the first occurrence of snoring in patients with OSA, and then stratified into two groups based on the differences between pre-S and S episodes: the increased group and the decreased group. HRV measurements during the pre-snoring (non-snoring) and snoring periods were accurately obtained using a reliable algorithm that can detect the exact onset of snoring. This visualizing algorithm of snoring revealed that the typical features of the clustered snores in the amplitude domains were near-isometric spikes (most had an ascending–descending trend) and had one or more visibly fixed frequency bands [19]. 

In the current study, two factors were identified as being associated with a higher probability of an increased LF/HF ratio during the initial snoring episode. One factor was patients experiencing their first snoring episode after lying down to sleep for more than 20 min. This finding suggests that patients may experience increased sympathetic activity after lying down to sleep for more than 20 min, as indicated by the elevated LF/HF ratio during the initial snoring episode. This challenges the common belief that snoring elevates sympathetic nervous system activity. Interestingly, the observations revealed that sympathetic activity before sleep was initially high and gradually decreased to a baseline level after more than 20 min of lying down. This implies that if snoring occurs within the first 20 min of sleep and the sympathetic activity has not yet returned to baseline, it could mistakenly be assumed that snoring causes a decrease in sympathetic activity compared to the pre-snoring period. For individuals experiencing initial snoring beyond the first 20 min of sleep onset, heightened vigilance and monitoring of their sleep patterns are recommended. This is important because early snoring could indicate underlying sleep-disordered breathing, requiring further evaluation and appropriate interventions to improve sleep outcomes.

The other factor associated with a higher probability of an increased LF/HF ratio during the initial snoring episode was patients with severe OSA. OSA is known to be accompanied by elevated sympathetic drive, likely due to an increased chemoreflex drive even during periods of normal oxygen levels. Furthermore, conditions such as short sleep duration, low sleep efficiency, and insomnia, when combined with short sleep duration, were found to be associated with higher levels of sympathetic tone [28]. Another study suggested that targeted interventions aimed at enhancing the functioning of the ANS could have positive effects on sleep quality [29]. Considering the well-established association between OSA and cardiovascular risks, these findings suggest that monitoring ANS activity, as reflected by LF/HF ratios, may offer valuable insights into the cardiovascular health of OSA patients. Such information could guide focused therapeutic interventions aimed at improving autonomic function and potentially reducing the adverse cardiovascular effects associated with severe OSA. In the current study, based on 5 and 10 min epoch analysis, parasympathetic activity, measured by RMSSD, significantly increased during the first hour of non-REM sleep. In Burgess’s study, cardiac parasympathetic activity, measured by HF, showed minimal variation between sleep stages [30]. Furthermore, in Burgess’s study, cardiac sympathetic activity (pre-ejection period, PEP) decreased linearly during sleep. Additionally, in the current study, sympathetic activity, measured by the LF/HF ratio, decreased within the first 20 min of sleep and then exhibited individualized oscillations.

Some studies have examined the impact of CPAP therapy on sympathetic activity in patients with OSA. A review study found that sympathetic activity increased during sleep in adult patients with OSA [31]. CPAP is typically regarded as the primary treatment for OSA. However, evidence-based clinical guidelines recommend the use of oral appliances (OAs), including mandibular advancement devices, for OSA if CPAP therapy proves ineffective or if a patient prefers OA treatment, regardless of the severity of OSA [32]. A previous study showed that the apnea/hypopnea index (AHI) and oxygen desaturation index (ODI) values improved after three months of mandibular advancement device (MAD) treatment [33]. A review study reported that increases in sympathetic activity could be attenuated by treatment with continuous positive airway pressure (CPAP), which mediated the amelioration of apneic events [34]. Another systematic review and meta-analysis reported that patients with OSA who received treatment with CPAP experienced a reduction in the burst frequency of muscle sympathetic nerve activity [35]. An earlier study reported that long-term (6–12 months) CPAP treatment reduces muscle sympathetic nerve activity in patients with OSA [36]. Another study reported that cardiac variability improves as an acute effect on the first night of CPAP use, especially during the non-REM stage, in patients with severe OSA [37].

However, there are relatively limited data regarding the impact of Oral Appliance (OA) therapy for OSA on sympathetic activity. A previous study reported that patients with OSA experienced a reduction in the LF/HF ratio after three months of treatment with an oral jaw-positioning appliance [38]. Another study reported that there were no differences in daytime cardiac autonomic function changes between MAD and CPAP treatments in patients with OSA [39].

The strength of this study lies in the use of a reliable algorithm that visualizes the typical features of snoring frequency and amplitude, allowing for the exact snoring segment can be calculated, and visualized on a spectrogram [19]. In this study, this reliable algorithm captured the exact timing of the initial snoring, followed by the calculation of the heart rate variability (HRV) parameters during the first hour of sleep accordingly. However, this current study has several limitations. Firstly, a small number of participants were enrolled in the current study (n = 45). This is due to the strict inclusion criteria that required participants to have previously been referred for PSG and diagnosed with OSA (AHI ≥ 5). As a result, only a limited number of participants were included in this study. Secondly, although the recordings were obtained during whole-night PSG, the HRV measurement was only analyzed during the first hour of sleep (non-REM sleep). Therefore, the sympathetic and parasympathetic tones may differ between non-REM sleep and REM sleep. Further studies with larger sample sizes and longer durations of HRV measurements are needed to confirm the findings.

## 5. Conclusions

In the current study, an increase in the value of RMSSD and a decrease in the value of the LF/HF ratio were observed during the first hour of sleep for patients with OSA. Additionally, there was a higher probability of an increased LF/HF ratio (indicating sympathetic function) in initial snoring occurring more than 20 min after lying down to sleep and in patients with severe OSA.

## Figures and Tables

**Figure 1 healthcare-11-02701-f001:**
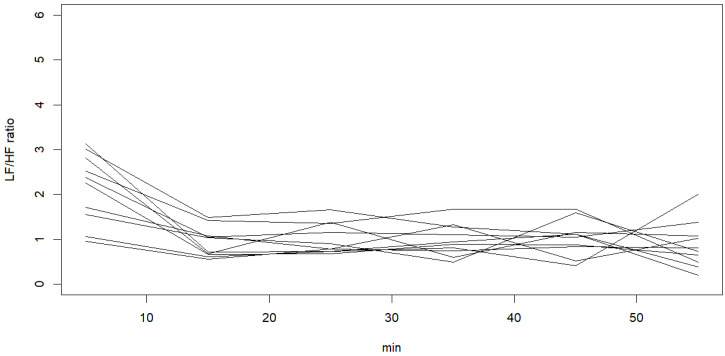
Ten randomly selected individual values of LF/HF ratio within the first hour of sleep.

**Figure 2 healthcare-11-02701-f002:**
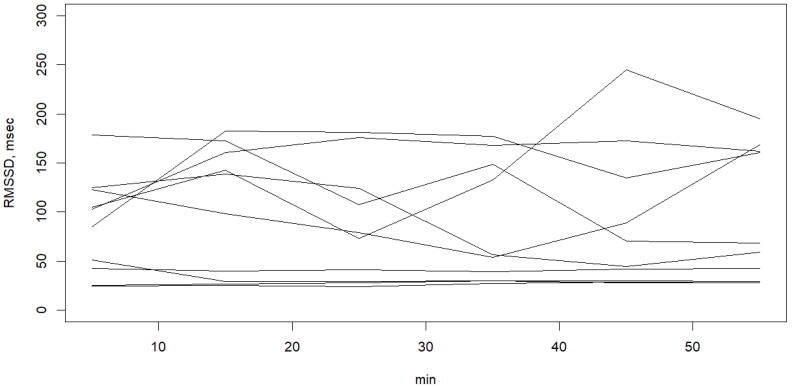
Ten randomly selected individual values of RMSSD within the first hour of sleep.

**Figure 3 healthcare-11-02701-f003:**
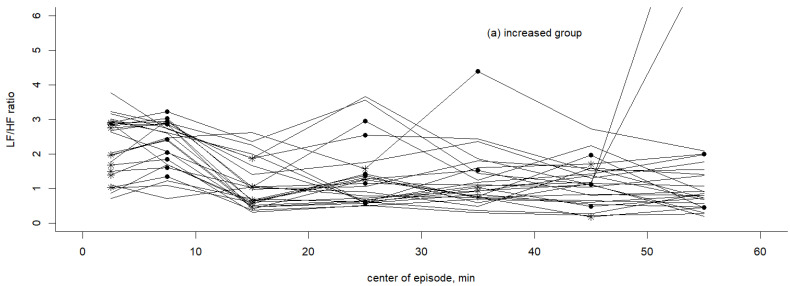
(**a**) Increased group and (**b**) Decreased group for LF/HF ratios from pre-S episodes to S episodes during the first hour of sleep. Star points: centers of pre-S episodes. Black round points: centers of S episodes.

**Table 1 healthcare-11-02701-t001:** Demographic characteristics of participants.

	Total	Mild–Moderate: 5 < AHI ≤ 30	Severe: AHI > 30	T or W or χ^2^	df	*p*
Number	45	27	18			
Age, years	43.8 ± 11.9	45.4 ± 12.8	41.5 ± 10.4	1.080	43	0.286
BMI, kg/m^2^	29.1 ± 5.4	27.6 ± 4.1	31.3 ± 6.4	−2.174	26.3	0.039
Neck circumference, cm	40.0 (38.0, 43.0)	39.0 (37.0, 41.5)	42.0 (40.0, 45.5)	116.5		0.003
AHI	20.6 (9.1, 56.6)	10.6 (7.6, 19.4)	62.3 (41.2, 73.6)	0		<0.001
Center of pre-S episode, min	7.5 (2.5, 15.0)	15.0 (7.5, 15.0)	2.5 (2.5, 13.1)	341		0.019
Center of S episode, min	15.0 (7.5, 25.0)	25.0 (15.0, 35.0)	7.5 (7.5, 22.5)	344.5		0.016
Center of S episode within 10 min, yes vs. no	17/28	5/22	12/6	10.6	1	0.001
Center of S episode within 30 min, yes vs. no	34/11	19/8	15/3	1.0	1	0.322
Pre-S episode LF/HF ratio	1.83 ± 1.00	1.9 ± 1.1	1.8 ± 0.9	0.4	43	0.712
Pre-S episode RMSSD, msec	39.8 (26.8, 72.2)	35.8 (26.6, 63.3)	59.7 (28.4, 72.1)	202		0.354
S episode LF/HF ratio	1.77 ± 0.99	1.61 ± 0.99	1.99 ± 0.97	−1.3	43	0.212
S episode RMSSD, msec	46.5 (29.7, 101.3)	39.6 (29.8, 71.8)	67.2 (30.8, 121.9)	197		0.295

Abbreviations: AHI: apnea/hypopnea index; BMI: body mass index; HF: absolute power of the high-frequency band (0.15–0.4 Hz); LF: absolute power of the low-frequency band (0.04–0.15 Hz); RMSSD: root mean square of successive differences.

**Table 2 healthcare-11-02701-t002:** The mean HRV values stratified by recording time for the first hour of lying in bed.

	0–5 min	5–10 min	10–20 min	20–30 min	30–40 min	40–50 min	50–60 min	Df	F	*p*
LF/HF ratio	2.40 ± 0.78	2.29 ± 0.65	1.15 ± 0.85	1.20 ± 0.85	1.13 ± 0.76	1.07 ± 0.59	1.87 ± 2.49	1	13.19	<0.001
RMSSD, msec	62.17 ± 51.37	60.81 ± 49.84	67.48 ± 50.65	70.90 ± 52.53	71.73 ± 52.24	78.66 ± 58.20	80.41 ± 60.31	1	5.12	0.024

Abbreviations: HF: absolute power of the high-frequency band (0.15–0.4 Hz); HRV: heart rate variability; LF: absolute power of the low-frequency band (0.04–0.15 Hz); RMSSD: root mean square of successive differences.

**Table 3 healthcare-11-02701-t003:** The change of LF/HF ratio from pre-S episode to S episode group.

Variables	Total	Increased Group	Decreased Group	t or W	df	*p* Value
N (female/male)	45 (7/28)	25 (3/22)	20 (4/16)			0.682
AHI						0.359
Mild–moderate, 5 < AHI ≤ 30	27 (60%)	13 (52%)	14 (70%)			
Severe, AHI > 30	18 (40%)	12 (48%)	6 (30%)			
Center of pre-S episode, min	7.5 (2.5, 15.0)	15 (2.5, 25)	7.5 (2.5, 7.5)	186.5		0.134
Center of S episode, min	15.0 (7.5, 25.0)	25.0 (7.5, 35.0)	15.0 (7.5, 15.0)	197		0.216
t		−11.95	−6.69			
df		24	19			
*p* value		<0.001	<0.001			
Pre-S episode LF/HF ratio	1.8 ± 1.0	1.4 ± 0.8	2.4 ± 1.0	3.656	43	0.001
S episode LF/HF ratio	1.8 ± 1.0	2.0 ± 1.0	1.5 ± 0.9	−1.68	43	0.100
t		−4.45	5.53			
df		24	19			
*p* value		<0.001	<0.001			

Abbreviations: AHI: apnea/hypopnea index; HF: absolute power of the high-frequency band (0.15–0.4 Hz); LF: absolute power of the low-frequency band (0.04–0.15 Hz).

**Table 4 healthcare-11-02701-t004:** Factor associated with increased LF/HF ratio in initial snoring episode by logistic regression analysis.

	O.R.	Estimate	S.E.	z	*p*
Snoring after lying down more than 20 min	10.9	2.39	0.82	2.90	0.004
Severe OSA	5.01	1.61	0.80	2.01	0.045
Intercept		−1.35	0.65	−2.1	0.037

The modified Hosmer–Lemeshow F test was passed (*p* = 0.83). The area under the ROC curve = 0.779 with 95% CI: 0.642–0.916. Abbreviations: HF: absolute power of the high-frequency band (0.15–0.4 Hz); LF: absolute power of the low-frequency band (0.04–0.15 Hz).

## Data Availability

The datasets generated during and/or analyzed during the current study are not publicly available, but are available from the corresponding author (Y.-H.K.) on reasonable request.

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
