# Peer review of "Oscillation of Sympathetic Activity in Patients with Obstructive Sleep Apnea during the First Hour of Sleep"

_healthcare, 2023, doi:10.3390/healthcare11192701_

Round 1

Reviewer 1 Report

The authors presented a paper about Sympathetic Activity in Patients with 2 Obstructive Sleep Apnea During the First Hour of Sleep.

The paper is interesting but needs some modifications.

Authors must include the power analysis for sample size patients calculation.

Authors must change the english style not using "we" or "our" in the paper. 

Many abbreviation must be clarified in the paper. 

Author must include in the introduction references as follow: 10.4172/2167-0277.1000283,  10.3390/app10228175

Authors must change the english style not using "we" or "our" in the paper. 

Author Response

Dear Reviewer,
Please see the attached word file for our sincere reply.
Thanks again for your kind review!

Reviewer 2 Report

In this manuscript (healthcare-2599765), the authors investigated the effects of sleep and snoring, and contributors on sympathetic activity in patients with OSA. They concluded that higher LF/HF ratios were associated with the first occurrence of snoring while lying down for more than 20 minutes and with patients with severe OSA, but I have some concerns about this MS:

The authors mentioned that the HRV measurement was analyzed only during the first hour of sleep in the limitations. If such a method of analysis was applied or the variability was already shown in a previous study, please cite the literature. In the absence of the literature, the authors should prove the validity of the method of analysis for the first hour of sleep.

There were some parts of similar explanations repeated. For example, the authors excluded 9 participants of 57, but they repeated the description both in the methods and results. Similarly, the descriptions of the definition of “S episode” and “pre-S episode” and analyzing HRV at different time intervals after lying down are also repeated. These similar explanations in results can be omitted.

In the discussions, the authors suggested that patients may experience increased sympathetic activity after lying down to sleep for more than 20 minutes and the activity gradually decreases to a baseline level after more than 20 minutes of lying down. On the other hand, the activity has not returned to baseline in case of snoring occurrence within the first 20 minutes. So, they recommend monitoring the sleep pattern in patients who experience initial snoring beyond the first 20 minutes of sleep.

These passages are confusing, and it is kinder for readers to show these arguments around the 20 minutes in a new schematic illustration or add the baseline and following data to Figure 3.

Please add discussions about the possible impacts of CPAP or oral appliance therapies on sympathetic activity.

Author Response

(The authors gave the same response as above.)

Reviewer 3 Report

The authors aim to investigate the combined effect of sleep and snoring on sympathetic activity and explore factors that might contribute to increased sympathetic activity in individuals with OSA during the first hour of sleep. It remains unclear why the study was limited to the analysis of the first hour of sleep because the authors explained that their focus will be on non-REM sleep. Polysomonographic data were availaible so that nonREM-sleep phases of whole night recordings could have been used instead. The data analysis also had the aim to explore factors that might contribute to increased sympathetic activity in individuals with OSA. For this purpose the heart rate variability (HRV) analysis was used to assess autonomic nervous system function.

The results are presented in a good scientific standard. Tables and figures are appropriate to illustrate the results. But the results are difficult to interpret. This finding suggests that patients may experience increased sympathetic ctivity after lying down to sleep for more than 20 minutes, as indicated by the elevated LF/HF ratio during the initial snoring episode. This challenges the common belief that snoring elevates sympathetic nervous system activity. Interestingly, our observations revealed that sympathetic activity before sleep was initially high and gradually decreased to a baseline level after more than 20 minutes of lying down. This implies that if snoring occurs within the first 20 minutes of sleep and the sympathetic activity has not yet returned to baseline, it could mistakenly be assumed that snoring causes a decrease in sympathetic activity compared to the pre-snoring period.

The authors cannot live up to their initial claim of analyzing the combined effect of non-REM sleep and snoring. In their interpretation of the data it is not defined how the baseline sympathetic activity was assessed. A comparison to other studies with focus on sympathetic activity during sleep would be helpful, e.g. Burgess et al. (2004) who investigated the course of the pre ejection phase (PEP) of the left ventricle as a marker of sympathetic activity during whole-night polysomnographic studies, including the observation of the PEP during REM sleep.

Author Response

(The authors gave the same response as above.)

Round 2

Reviewer 2 Report

In this manuscript (healthcare-2599765), the authors revised the first version.

These revisions may be accessible to readers, but it is better to add a supplementary explanation for Figure 3 in Results.

Author Response

(The authors gave the same response as above.)

Reviewer 3 Report

The reviewer's suggestions were taken seriously and implemented very well.

Author Response

(The authors gave the same response as above.)
